

# Coastline evolution based on statistical analysis and modelling

Elvira Armenio[1], Francesca De Serio[1], Michele Mossa[1], Antonio F. Petrillo[1]

Department of Civil, Environmental, Land, Building Engineering and Chemistry (DICATECh), Polytechnic University of Bari, Italy

*Correspondence to*: Elvira Armenio (elvira.armenio@poliba.it)

**Abstract.** Wind, waves, tides, sediment supply, changes in relative sea level and human activities strongly affect shorelines, which constantly move in response to these processes, over a variety of time scales. Thus, the implementation of sound coastal zone management strategies needs reliable information on erosion and/or deposition processes. Suggesting a feasible way to provide such information is the main motivation of the present work. A chain approach is here proposed, tested on a vulnerable

coastal site located along the southern Italy, and based on the joint analysis of field data, statistical tools and numerical modelling. Firstly, the coastline morphology has been examined through interannual field data, such as aerial photographs, plane-bathymetric surveys, seabed characterization. After this, rates of shoreline changes have been quantified with a specific GIS tool. Correlations among the historical shoreline positions have been detected by statistical analysis and have been satisfactorily confirmed by numerical modelling, in terms of recurrent erosion/accretion area and beach rotation trends. Finally,

based on field topographic, sediment, wave and wind data, the response of the beach by the numerical simulation has been investigated in a forecasting perspective. The scope of this study is providing a feasible, general and replicable chain approach, which could help to thoroughly understand the dynamics of a coastal system, identifying typical and recurrent erosion/accretion processes, and predict possible future trends, useful for planning coastal activities.

## 1 Introduction

Beachfront lands are the place where unique and fragile natural ecosystems evolve in equilibrium with the ever-changing forces of wind, waves, and water levels. Even if highly vulnerable to natural hazards including marine inundation, floods, storm impacts, sea-level rise, and coastal erosion, these coastal areas are the site of intense residential and commercial development, thus being even more vulnerable. Shoreline evolution, characterized by erosion and deposition areas, have consequences on socio-economic activities and ecosystems, therefore their evolving and understanding represents a challenge

to coastal communities, coastal infrastructures and adjacent estuarine environments (Cutter et al., 2008; Torresan et al., 2012; De Serio and Mossa, 2014; 2016; Samaras et al., 2016; Armenio et al., 2017; De Padova et al., 2017)

To evaluate changes in coastal regions and recognize some key physical processes over different historical timescales (decade to century), data of shoreline geometry and position are basic indicators. A quantitative analysis of data of shoreline evolution at different timescales and with a fine spatial resolution is fundamental in establishing the processes driving erosion and





accretion (Elfrink 1998; Katz et al, 2013; Thébaudeau et al., 2013; Oyedotun, 2014). Thus, various statistical methods of determining rates of shoreline change have been studied and applied (Dolan et al., 1991).

One of the simplest methods is the end-point rate (EPR) method, which estimates the distance of the shoreline movement rated by the time elapsed between the oldest and the most recent shoreline (Genz et al., 2006). Foster and Savage (1989) used the

average of rates (AOR) method, which computes separate end-point rates for more than two combinations of shorelines. The linear regression rate-of-change (LRR) statistic has been used by fitting a least squares regression line to all shoreline points for a transect (Dolan et al., 1977), thus deducing the rate as the slope of the line. An iterative linear regression fitting all possible combinations of shoreline points, leaving out one point in each iteration, has also been implemented, i.e. the jackknife (JK) method (Dolan et al., 1991). In addition to the above written established methods, also the weighted linear regression

(WRL) method has been used (Genz et al., 2006). In this case, more reliable data is given greater emphasis, or weight, defined as a function of the variance in the uncertainty of the measurement.

Questions raise about the appropriateness of linear models, considering that shorelines do not recede or accrete in a uniform manner (Douglas et al., 2000, Thieler et al., 1994a, 1994b). As an example, coastal embayments featured by a parabolic curve, which are representative of more than 50% of the world's coastlines, are very dynamic environments where the shoreline

position can fluctuate significantly due to processes such as beach rotation (Short et al., 2004; Blossier et al., 2015). This can be defined as the landward or seaward movement at one end of the beach accompanied by the reverse pattern at the other end (Bryan et al., 2013) and is often a consequence of maritime constructions (i.e. dikes, breakwaters) and variations of river sediments supply on flanking beaches. As well, in shorter-term, also changes in wave direction could contribute to this marked shoreline readjustment.

Besides the statistical analysis of data, for the long-term shoreline evolution also the use of analytical, morphodynamic and physical models has been increasingly demanded (Deigaard et al., 1986a; Deigaard et al., 1986b; Dean and Dalrymple, 2004; Davidson et al., 1991; Thomas et al., 2013). Nevertheless, coastal morphodynamics models require large computational resources and time and, consequently, they are scarcely suitable to large spatial and temporal scales over which beaches evolve. Physical models are well suited to local analysis but are often prohibitive to be used for very large scales. This means that not

necessarily the increasing complexity of used models improves the predictions. Moreover, all models need to be calibrated and validated by means of sensitivity analyses, demanding for rich and complete sets of data.

In this work we aim to show that: i) the statistical analysis of data remains an accurate method to characterize shoreline changes, even if it disregards potential changes due to engineering activities or major climate change; ii) the use of a simple one-line numerical model, based on the conservation of sand volume equation, is still satisfactorily to evaluate shorelines changing,

with the advantage of being feasibly applicable. Therefore, the present paper proposes a chain approach to detect information on the shoreline evolution, based on statistical analysis and one-line modelling (Figure 1).

Firstly, field information and shoreline data have been analyzed to examine the past behavior of the coastal system and the effects of human activities on shoreline movement and rates of change. After this, GIS tools have been used for the quantification of shoreline rate of change for interannual periods. A regression model and the Person's correlation matrix have



been used to statistically investigate on possible correlations of historical shoreline profiles. Finally, the numerical model LITPACK developed by the DHI - Danish Hydraulic Institute (DHI, 2016), implemented with field data, has been validated with hindcasting and used for a short-term shoreline prediction.

This approach has been applied to a target area located in southern Italy along the Adriatic Sea, characterized by a coastline 18 km long and described in Section 2. Section 3 illustrates the long-period and the fine resolution spatial data derived from the observations collected for this site. It also explains the principal features of the used GIS tool and LITPACK model. The quantitative analysis of shoreline changes in space and time is presented in Section 4, while Section 5 shows the results of the numerical simulation, both in hindcasting and in forecasting terms. The presented results are site-specific but the used procedure is general, replicable, and applicable to similar data sets.

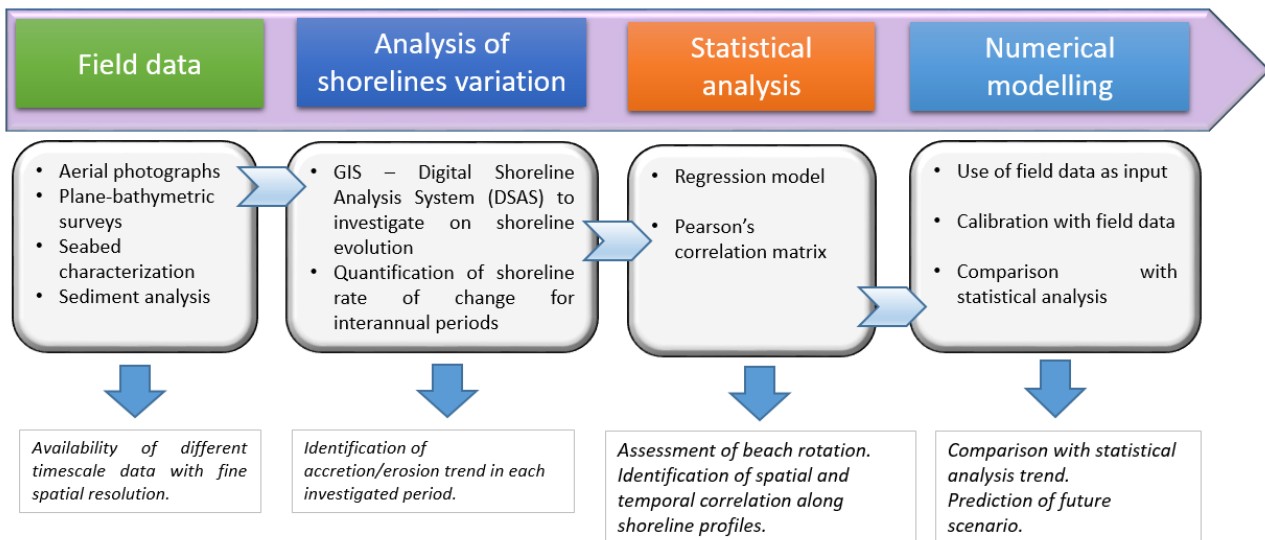

**Figure 1: Sketch of the proposed approach.**

## 2 Study site

The study area is in the Southeastern coast of the Apulian region (Italy) along the Adriatic Sea, namely in the gulf of Manfredonia. It extends from Margherita di Savoia town to Barletta town, with a total length of about 18 km (Figure 2). The coast here is typically a low sandy beach with dunes, wetlands and salt marshes. At approximately 2 km off the coast the depth is around 13 m. This sandy coast has originated over the years from the sediments supplied by several rivers, flowing into the gulf. The coast neighboring Margherita di Savoia is mainly due to the solid transport contribution of the most important river of the region, the Ofanto river, whose length and flow rate are respectively 134 km and 15 m³/s (annual average).

In the last two centuries, both the rivers and the coastal area have experienced remarkable transformations especially due to strong human activity, with consequent alternating erosion and deposition processes. In the early 1800's during some



remediation works, river sediments were used to bury marshes and in canalization works, thus provoking a reduction in sediment supply from land and a widespread coastal erosion. In the mid-1900's, several reservoirs and crossbars were constructed on the Ofanto river and its tributaries, to assure water supply for irrigation, industrial and drinkable uses. Since 1960, the intense urbanization of the coastal zone has provoked critical local issues, furtherly contributing to erosion

phenomena. Probably the most perturbing cause in the coastal dynamics between the Ofanto's mouth and Manfredonia town (Figure 2) was the construction of the port of Margherita di Savoia, started in 1952 and completed forty years later. Intercepting the coastal flow rich of sediments, this structure has always had a great impact on the adjacent coast, altering the beach equilibrium and inducing over the years localized heavy erosion and accumulation, thus leading to a change in the coastal morphology from linear to curve shape beach profile (Damiani et al., 2003).

This coastal sector is subjected to predominant NNW and SSE winds and the annual wave climate is characterized by a bimodal regime with a clear predominance of waves from N-NNE and E-ESE (Apulian Coastal Plan, 2012). The maximum significative wave height is in the range 1÷2 m, while the most frequent one is in the range 0.5÷1 m.

For the aim of the present work, the coastline in the study area has been divided into three parts with relatively homogeneous geomorphological change patterns, respectively named Cell I, Cell II and Cell III (Figure 2). They all have a curvy geometry.

Cell I and Cell III are two concave beaches (i.e. curved towards the sea) and are separated by Cell II, which is convex (i.e. curved towards the inland). Cell I is delimited by the Margherita di Savoia's port northerly and by the Ofanto river's mouth southerly. It has a length of about 6.0 km. Cell II extends from the Ofanto river's mouth up to a residential area called Fiumara, for a total length of about 1.5 km (Figure 1). Its convex coastline is characterized by the alternation of sandy and rocky beaches, with breakwaters and rip-rap seawalls also placed to protect the beach. Cell III connects the Fiumara site with the Barletta's

port, with a total length of about 8.6 km of sandy beaches.

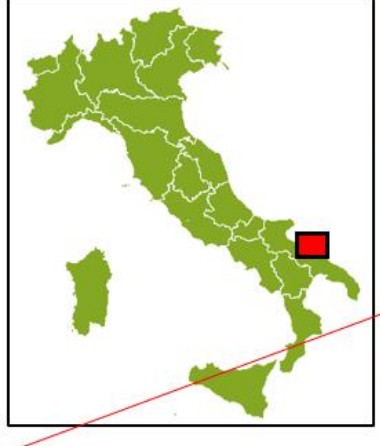
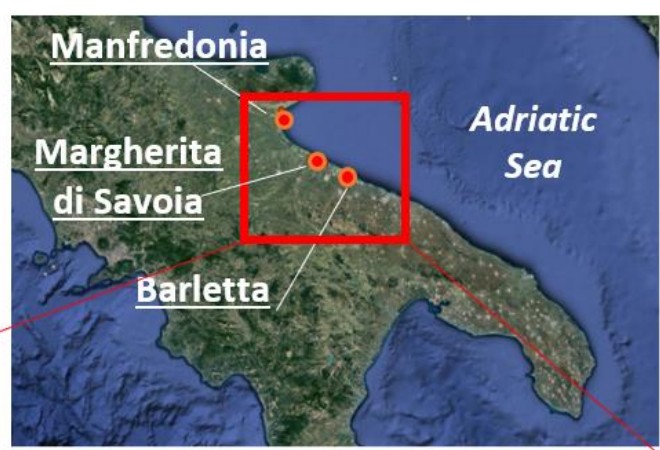

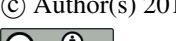



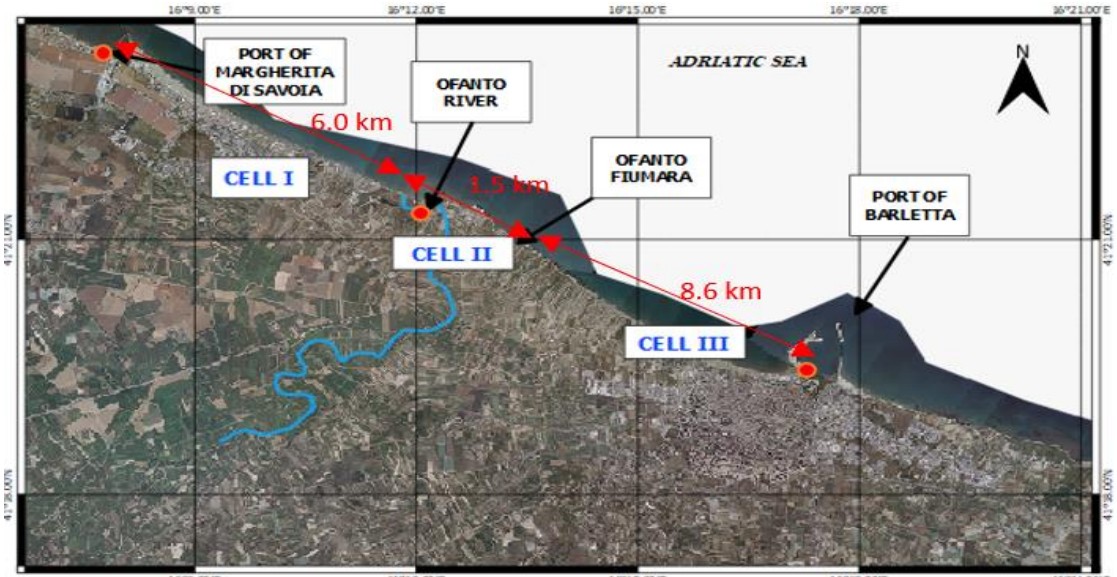

**Figure 2: Study area with notation of Cell I, Cell II and Cell III. Source Google Earth.**

## 3 Materials and methods

### 3.1 Availability of aerial and land data

To detect and quantify an erosion/accretion phenomenon, long-time observations are necessary in analyzing the evolution of the shoreline, thus to eliminate the influence of seasonal, episodic events such as individual storm surges and local sedimentary dynamics. For the present study, interannual field observations are adopted to map the historic coastline configurations. Shoreline positions have been derived from aerial photographs, digital orthophotos and global positioning-system field surveys.

As well known, the idealized definition of shoreline is that it coincides with the physical interface of land and water, but this definition is in practice a challenge to apply. The most common shoreline detection technique applied to visibly discernible shoreline features is manual visual interpretation, either in the field or from aerial photography (Boak and Turner, 2005). With aerial photography, the image has been corrected for distortions and then geo-referenced and adjusted to the correct scale, thus the shoreline has been digitized. In the field, a GPS has been used to digitize the visible shoreline feature in situ, as determined by the operator. Attention has been paid to ensure an accurate digitization and a critical review of the source materials. In fact, possible approximations could be due to difficulties in the interpretation of aerial photographs because of waves, swimmers and boats, or in geo-referencing aerial images because of wrong reference points. Consequently, we have assumed negligible the gaps between two shorelines in the range ± 3m (Chiaradia et al., 2008). This accuracy is important, considering that the



calculated measures of change obtained by DSAS are only as reliable as the sampling and measurement accuracy associated with the source materials (Oyedotun at al., 2014). Data have been digitized and appropriately overlapped for comparison relatively to the years 1992, 1997, 2006, 2008, 2011 and 2013.

Closer to the shoreline, fine resolution data assessed during a bathymetric survey performed in 2006 have been added. A plane-bathymetric survey and sediments analysis carried out in 2009 has also provided information on the nature of the seabed. A total of 15 on-shore samples have been collected at 5 cross-shore beach profiles along the target area between depths 1.73÷6.12 m. They are mostly composed by sand with a mean diameter in the range 1.67÷2.21 mm (Apulian Coastal Plan, 2012).

## 3.2 GIS application

In recent years, the Geographic Information Systems (GIS) technology has been used to create high-quality maps, visualize and simplify large data. Specifically, to quantify the coastal evolution in the investigated period, the shoreline variation has been statistically analyzed using the Digital Shoreline Analysis System (DSAS) extension in (ESRI) ArcGIS © software. The DSAS has been massively used in measuring, quantifying, calculating and monitoring shoreline rate of change statistics (see among others Brooks and Spencer, 2010; González-Villanueva et al., 2013; Jabaloy-Sánchez et al., 2014; Young et al., 2014). One of its main benefits in coastal change detection is its ability to compute the rate-of change statistics for a time series of shoreline vector data (Oyedotun at al., 2014, Thieler at al., 2009), together with the statistical data necessary to estimate the reliability of the calculated results. Among the many statistical options proposed by DSAS to analyze shoreline change data, including as an example EPR and LRR methods, the Net Shoreline Movement (NSM) method has been used in our study. That is, the DSAS has firstly been implemented to map the shoreline positions occurred during the investigated period, based on the available spatial data (e.g. maps, aerial photographs). Secondly, several transects orthogonal to the coastal orientation have been considered. The intersection between each transect with the historical shorelines has been marked and the distance between the oldest and the most recent shoreline has been computed. The distance migration of the shoreline, either seaward or landward, has been estimated for the period from 1992 to 2013.

## 3.3 LITPACK numerical model

The one-line model used in this work is the software package LITPACK by DHI (DHI, 2016). The one-line concept assumes that the beach profile shape (i.e. the cross-shore profile) remains unchanged as it advances or retreats, so that volume change is directly related to shoreline change (Frey et al. 2012). Spatial and temporal variations in longshore transport drive shoreline accretion or erosion. The LITPACK reproduces the littoral transport of non-cohesive sediment under the action of waves and currents, littoral drift, coastline evolution along quasi-uniform beaches (Deigaard et al., 1986; Fredsoe et al., 1985, Schoonees and Theron, 1995). Specifically, the LITPACK module *Coastline Evolution* has been utilized in the present application. The





development of the coastline due to littoral transport has been computed in turn from wave statistics, sediment properties and coastline configuration.

The model has been initialized with the field data of bathymetry and sediments described in section3.1. The bathymetry data acquired during the survey of 2006 have been interpolated onto a fine mesh (Figure 3). The bathymetry map shows depth

contours parallel to the coastline in both Cell I and Cell III. In the central Cell II a less uniform bottom slope is noted, with some slightly convex contours. The coastline orientation with respect to North is around 120.4° for Cell I, 123.69° for Cell II and 125.79° for Cell III. Data collected during previous surveys (sect. 3.1) show that sediments in this region consist of sand from fine to medium. This information has also been used to implement the model, characterizing the coast at five cross-shore beach profiles (Figure 3), two located in Cell I, two in the Cell II, and one in Cell III.

The erosion/progress of the shoreline has been correlated in time with the wave energy impacting the shoreline. To this purpose, a wave hind-casting analysis for the study area has been previously carried out, using the European Centre for Medium-Range Weather Forecasts (ECMWF) model. Wave data have been processed from 1992 to 2013, providing that storm surges with higher intensity have N÷NNE and E incoming directions, while the highest frequencies of occurrence are noted from NNO and ESE. The derived wave with an equivalent energetic contribution used to initialize the model has a significant wave height

of 0.77 m, a wave direction of 47°N and a wave period of 4.23s. The LITPACK model considers the Battjes and Janssen (1978) approach of wave propagation from deep water. For the present study no currents have been included in the simulation.

Numerical simulations have been performed for all the three Cells, referring to years 2005, 2008, 2011 and 2013 and the model has been validated based on the available field data of coastline changes. After this, the model has predicted the shoreline evolution up to year 2018.



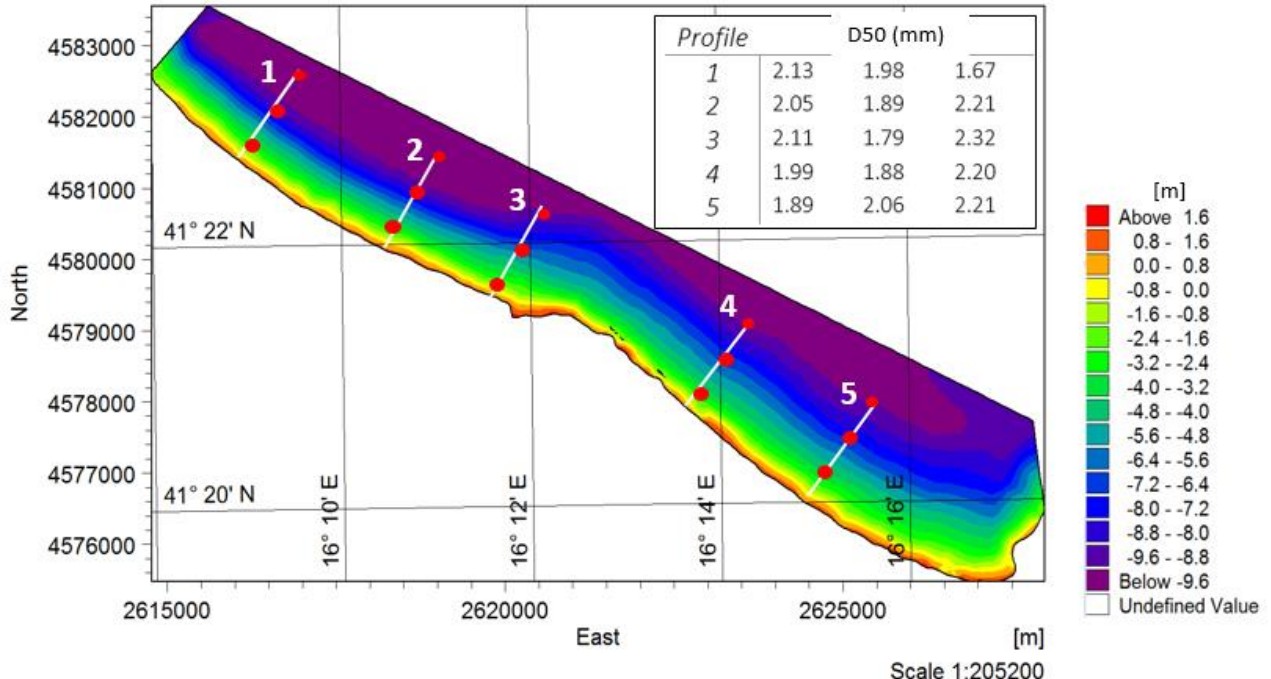

**Figure 3: Bathymetry of the study area (Gauss-Boaga coordinates) and cross-shore model profiles. Mean diameters (D50) measured in three locations along each profile are reported (from the coast to open sea).**

## 4 Statistical analysis results and discussion

Overlaying the historical shorelines of years 1992, 1997, 2005, 2008, 2011 and 2013, the first comparative spatial analysis has been executed, to analyze and map areas of accretion and erosion in all the investigated Cells. The value of the NSM with DSAS has been returned for equally spaced transects, representing the distance between the most recent and the oldest of the two compared coastlines. A total of 577 transects, each separated by approximately 25 m, have been superimposed on the study area: 243 transects in Cell I, 44 transects in Cell II and 289 transects in Cell III. The DSAS has been applied to five time-intervals for a: 1992-1997, 1997-2005, 2005-2008, 2008-2011 and 2011-2013. The computed shoreline rate of change has produced the following results.

### 4.1 Results in Cell I

Figure 4 shows that, during the years 1992-1997, in Cell I accumulation occurs in the northern area, from transect 0 to transect 110. In the central part, from transect 110 to 190, the shoreline remains quite stable. Conversely, erosion characterizes the southern area, from transect 190 to 244. During the period 1997-2005, both accumulation at North (transects from 0 to 150) and erosion at South (transect from 175 to 250) increase. Totally, in the time interval 1992-2005, there is an accretion of about



40 m in the northern shore and an erosion of about 60 m in the southern one. The trend reverses in the successive period 2005-2008: northerly, the accretion area experiences erosion returning to values of 1997, while the erosion area experiences strong accretion southerly. During years 2008-2011, the northern area shows stable condition with average shoreline changes around zero (from transect 0 to 175) and the southern area shows erosion. In this period, a rise of about 10 cm of the mean sea level

5  has been recorded, while in the successive period 2011-2013, a decrease in the average sea level of about 2 cm (Damiani et al., 2003) could justify the slight accretion characterizing both the northern and southern areas.

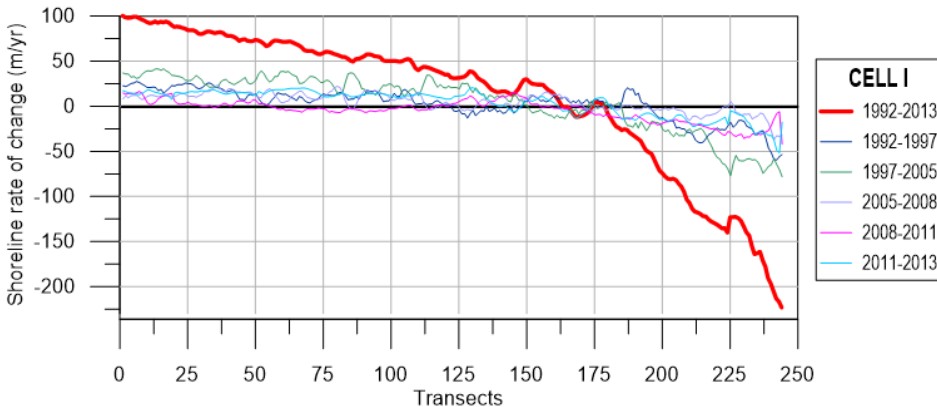

**Figure 4: Shoreline evolution during 1992-1997, 1997-2005, 2005-2008, 2008-2011, 2011-2013.**

Consequently, in the whole investigated period 1992-2013, two different areas can be recognized in Cell I (Figure 5): an advance area from transect 1 to 175, whose length is approximately 4300 m and a retreat area from transects 175 to 244, whose length is about 1700 m. The overall trend in Figure 5 clearly shows advance of about 100 m in the northern shore and retreat of about 200 m in the southern one. Further, the modified coastline keeps its concave shape even if it undergoes a clockwise

15  rotation. The rotation point is identified along transect 175, distant 4350 m from Margherita di Savoia port (Figure 2).



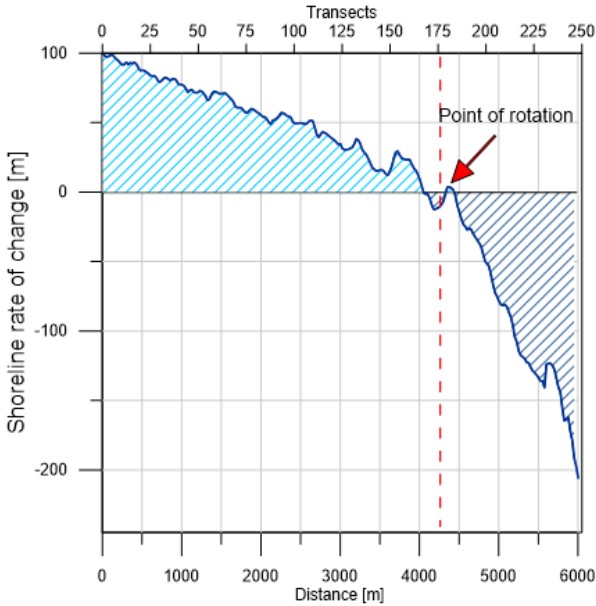

Figure 5: Shoreline changes between 1992 and 2013 at Cell I. (Note: light area = accretion, dark area = erosion).

To permit a thorough comparison in the different interannual periods, Cell I has been divided in three sectors (i.e. northern,

central and southern), each one enclosing the same number of transects. In the following plots (Fig. 6), these sectors are indicated specifying their delimiting transects (i.e. *Ti-Tj*, with *i* and *j* being the number of the first and last transect of that sector). For each interannual interval, the shoreline changes computed by DSAS along each transect and inside each sector have been averaged. The corresponding averaged values are plotted in Figure 6, with zero being the reference starting point (elder available shoreline position). Measuring the shoreline rate of change with respect to this zero-reference, positive values

mean accretion while negative values mean erosion. Nevertheless, this data representation allows us to estimate also the relative local trend, i.e. with respect to the previous and successive temporal interval shoreline position. Thus, it is possible to evaluate both an absolute and a relative accretion/erosion. It is evident that the southern shoreline experiences erosion from 1992 to 2013, with a maximum shore retreat in 1997-2005. In any case, a reduced retreat, corresponding to a local advance, is noted during the years 2005-2008. The central and northern shores always display accretion, during the time interval 1992-2013.

Diminished accretion is observed during 2005-2008 and 2008-2011, thus resulting in local erosion trends.



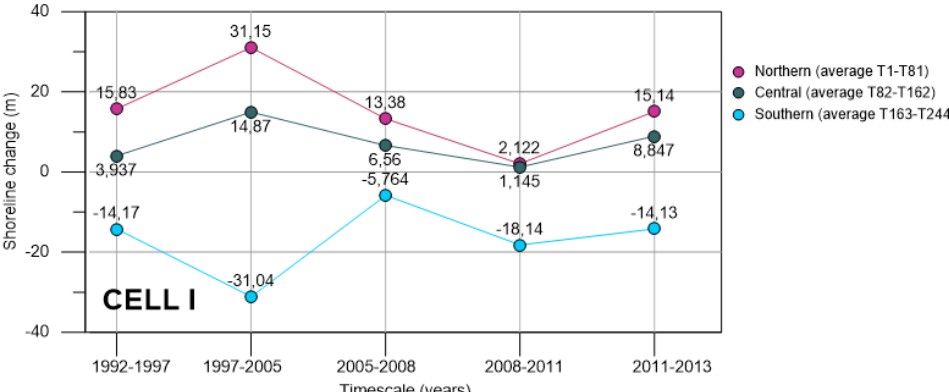

**Figure 6: Average shoreline position in northern, central and southern sectors of Cell I.**

The frequent shoreline variations observed have been mostly due to human intervention, which has modified the coast, altering the beach equilibrium over the years. In Cell I, with the construction of the Margherita di Savoia port in 1992, the southern pier has retained the sediments from the Ofanto river and transported them northward by long-shore currents, thus causing a remarkable advance of the shoreline.

## 4.2 Results in Cell II

In Cell II the shoreline evolution from 1992 to 2005 displays a progressive erosion at the Ofanto's mouth, more effective during the years 1997-2005, while a slight variation is noted southerly (Figure 7). From 2005 to 2013 an opposite tendency occurs with accretion around the Ofanto's mouth. The trend referred to the overall period 1992-2013 highlights that the shore has suffered a severe erosion near the river mouth, with a retreat of about 250m. It is worth noting that the erosive tendency decreases over time. This is not due to a reduced erosive action of waves and currents, but it is mainly due to physical changes in the Ofanto's mouth. In fact, after a deep erosive action between 1950 and 1992, because of a drastic reduction of the river solid transport, it has changed from a *delta* to an *estuary* configuration, thus it has become less erodible (Damiani et al., 2003). As for Cell I, also Cell II has been divided into a northern, a central and a southern sector, each one characterized by the same number of transects. The averaged shoreline changes for each sector and for the considered interannual periods are plotted in Figure 8. A steady erosion condition is evident in all the sectors for the entire observed period, especially in the time frame 1992-2005, with the highest shoreline retreat during the years 1997-2005 in the northern sector (-86.95 m), which corresponds to the Ofanto river's mouth. In the period 2005-2011 the northern and central shores are still in a condition of retreat with respect to the zero-reference, even if a process of advance establishes with reference to the previous temporal period. The analysis of the shoreline dynamics highlights that the eroded sediments in Cell II could transit through southern Cell I and finally deposit in the northern sector of Cell I.



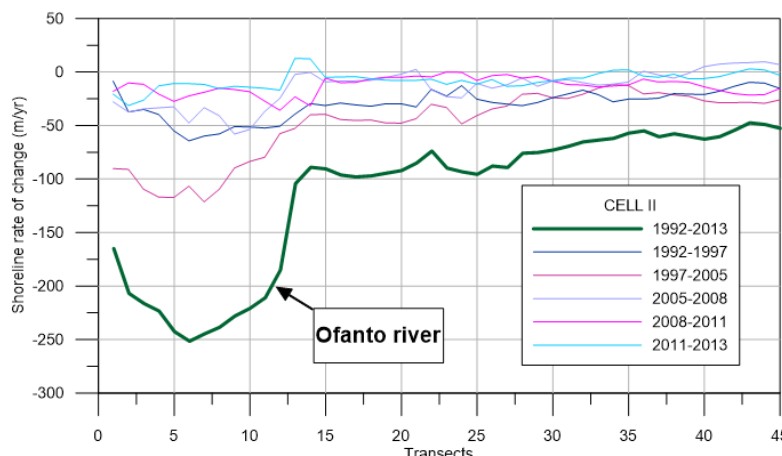

**Figure 7: Shoreline evolution during 1992-1997, 1997-2005, 2005-2008, 1992-2013, 2008-2011, 2011-2013.**

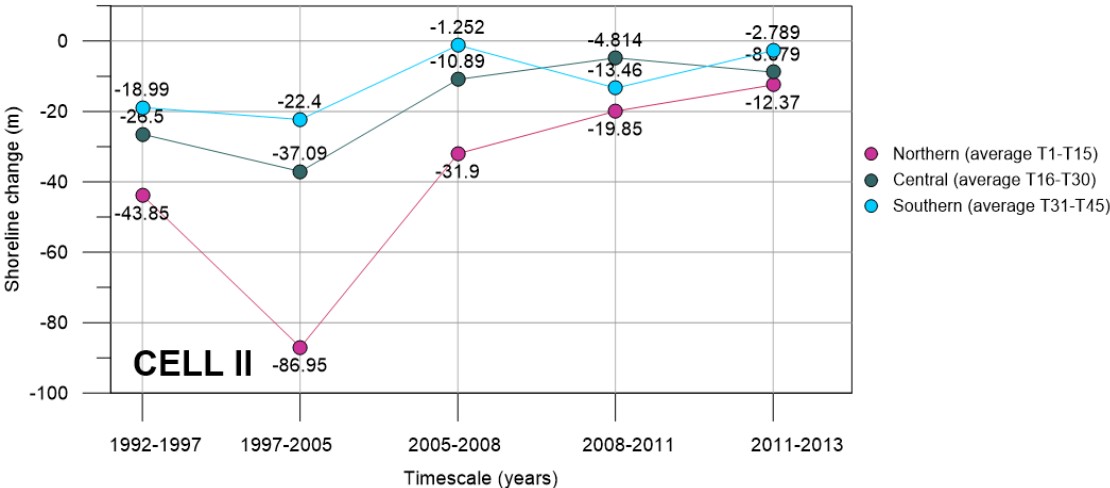

5    **Figure 8: Average shoreline position in northern, central and southern sectors of Cell II.**

### 4.3 Results in Cell III

The shoreline evolution of Cell III is displayed in Figure 9. In the period 1992-1997, a retreat occurs in the northern part (~ 10m), while southerly the shoreline remains quite stable. A similar tendency characterizes also the period 1997-2005, whit

10   stronger erosion northerly (~ 20m).

The successive period 2005-2011 shows an inverse tendency and accretion is noted especially in 2005-2008. The overall trend referring to the time frame 1992-2013 illustrates a substantial erosion experienced by the northern shore with an average retreat



in the shoreline position of about 30m. On the opposite, the southern shore shows a significant accumulation with an advance in the shoreline position of about 30m. This behavior is also synthetized in Figure 10. Namely, the shape of the northern shoreline modifies from quite linear (during 1992) to concave (during 2013). Analogously to the case of Cell I, a reversal point in advance/retreat can be recognized at about 3400 m from the northern limit of Cell III. In this case, an anticlockwise rotation

5 of the shoreline is argued (Fig. 10) and the coastline seems to evolve preserving its concave shape.

Based on the average rate of shoreline change in northern, central and southern sectors (Fig.11), in Cell III erosion with respect to the zero-reference is noted only along the northern sector, where a high landward excursion is evident from year 1992 up to year 2005. The central and southern shores are characterized by accretion, with exception of the period 2008-2011.

Similarly, to Cell I, in Cell III the construction of the Barletta's port has heavily modified the coastal dynamics, especially in

10 the northern region, determining an accumulation area. Overall, in the analyzed period, the sediments of Cell II have been transported both northwestward (to Cell I) and southeastward (to Cell III), leading to accretion areas near the two ports.

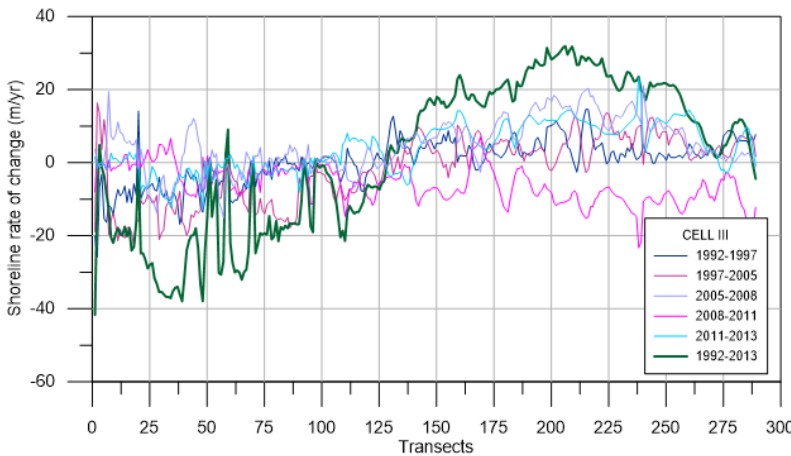

**Figure 9: Shoreline evolution during 1992-1997, 1997-2005, 2005-2008, 1992-2013, 2008-2011, 2011-2013.**



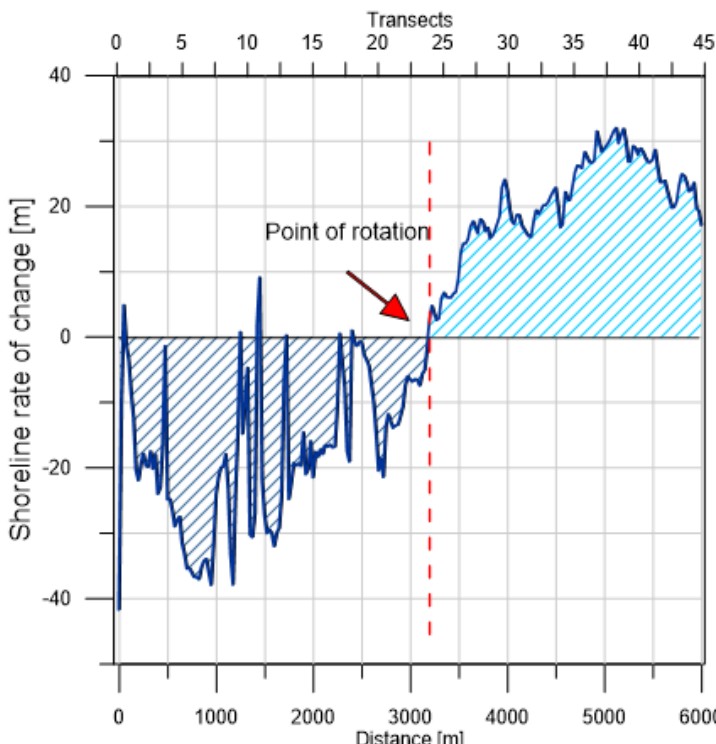

**Figure 10: Shoreline change between 1992 and 2013 at Cell III. (Note: light area = accretion, dark area = erosion).**

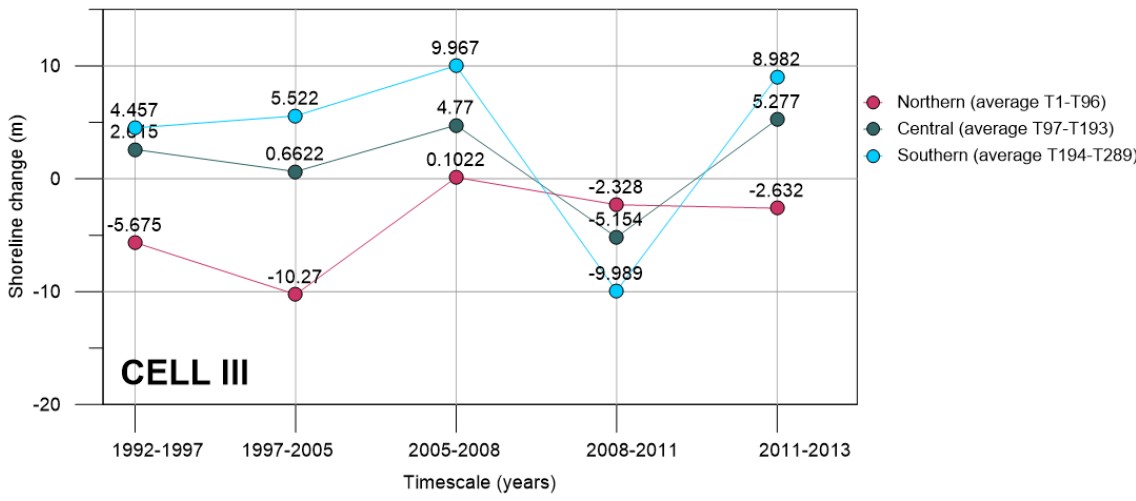

5 **Figure 11: Average shoreline position in northern, central and southern sectors of Cell III.**





### 4.4 Assessment of beach rotation by using regression analysis and Pearsons' matrix

Some useful information can be deduced by plotting in a joint graph the temporal shoreline changes occurred within each Cell (see Figure 12a, b, c), with reference to the southern, central and northern sectors respectively. A high correlation between the southern shoreline retreat and the northern shoreline advance is evident in Cell I (Fig. 12a) and is fitted by the linear regression

model $y = -80.427x + 170.77$ with a correlation coefficient $R^2 = 0.93$. In Cell II, the large erosion is proved by the linear

regression model $y = 68.013x - 250.02$ with $R^2 = 0.90$ (Fig. 12b). Also in Cell III a linear regression model (

$y = 19.869x - 37.637$ with $R^2 = 0.93$) expresses the shoreline behavior, in this case with opposite slope in comparison to

the linear regression model of Cell I, thus indicating retreat in the northern part and advance in the southern one (Fig. 12c).

A further step has been made, investigating in greater detail the mutual influence of each sector on the adjacent one.

Correlations between the northern-central, southern-central and southern-northern sectors are shown in Figure 13, respectively in the left, central and right columns, for Cell I (top row), Cell II (central row) and Cell III (bottom row). In each subplot the

regression equation is written, together with the corresponding $R^2$.

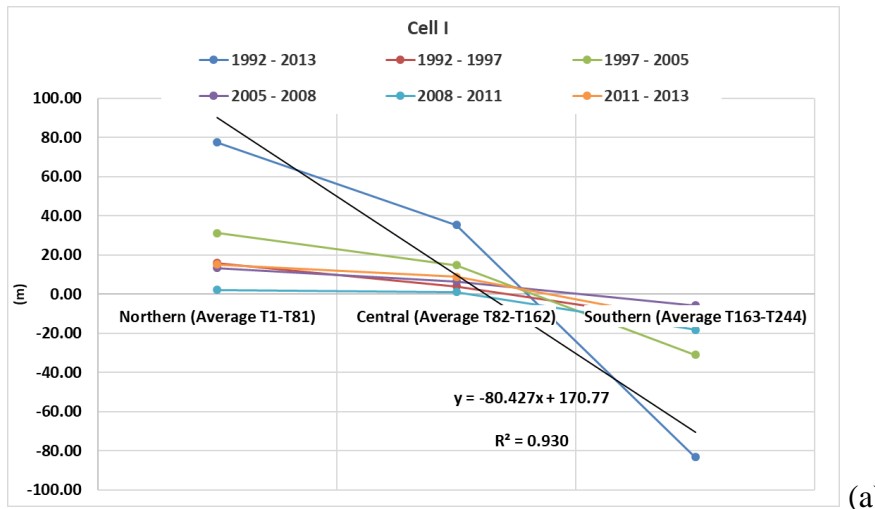

(a)




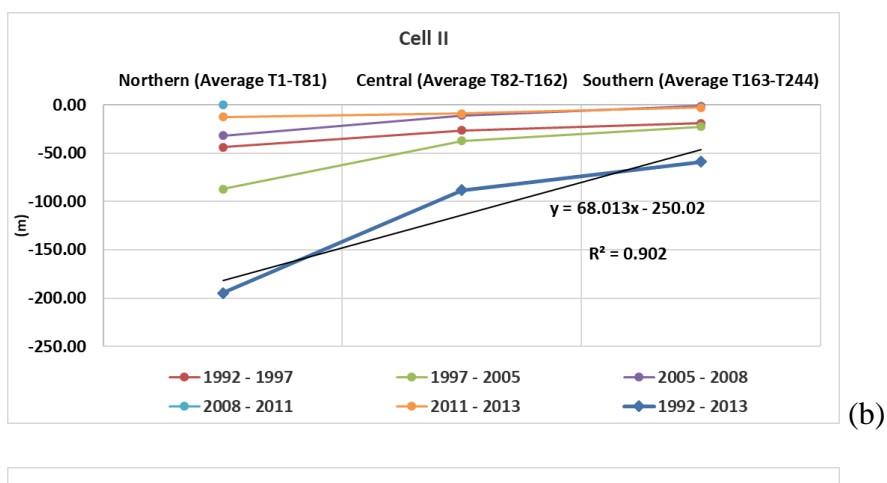

(b)

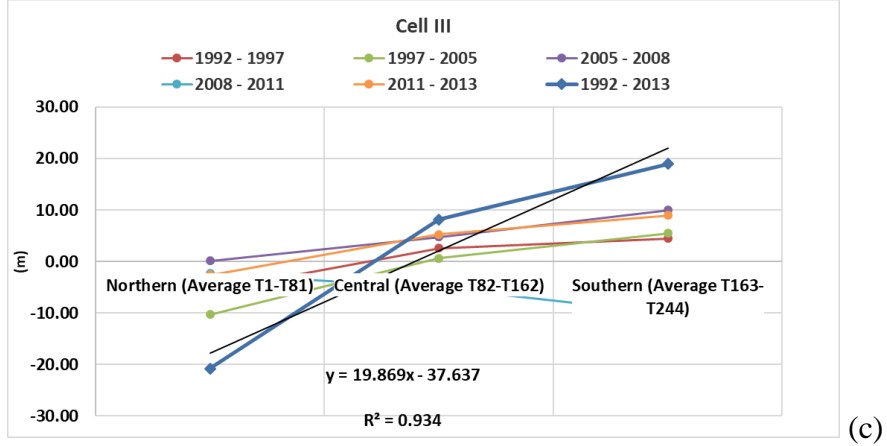

(c)

**Figure 12: Average variation (m) at northern, central and southern shore in (a) Cell I, (b) Cell II and (c) Cell III for the years 1992-2013, 1992-1997, 1997-2005 and 2005-2008.**

It is evident that in Cell I, when changes occur in the northern sector, they occur with the same sign in the central sector (Fig. 13a), while when changes occur in the southern sector, they occur with the opposite sign in the central sector (Fig. 13b). However, the statistically highest negative relation observed between the southern and northern sectors is the most interesting, proving a clockwise beach rotation (Fig. 13c). In Cell II, a positive relation is estimated for all the cases: between northern and

10 central sectors (Fig. 13d), southern and central sectors (Fig. 13e), and southern and northern sectors, confirming that in Cell II no beach rotation occurs (Fig. 13f). In Cell III a negative relation establishes between the northern and central sectors (Fig. 13g), as well as between the southern and northern sectors (Fig. 13i), while a positive relation establishes between the southern and central sectors (Fig. 13h). Even if in Cell III the $R^2$ coefficients are the lowest, due to a greater data scattering, in any way the examined trends explain a counterclockwise beach rotation.




To investigate thoroughly these linear correlations, the Pearson correlation coefficient, $r$, has been computed. Specifically, it provides a measure of the linear association between two continuous variables, in this case assumed to be some appropriate profiles, obtained in the following way. The 244 transects of Cell I have been grouped in 15 profiles (P1÷P15 from North to South). Similarly, the 45 transects of Cell II have been grouped in 9 profiles (P1÷P9 from North to South) and the 289 transects

5    of Cell III in 12 profiles (named P1÷P12 from North to South). Each of these profiles represents the time-average of the shoreline changes observed in the period 1992-2013 along few consecutives transects. As an example, the profile P1 is the time-average of the shoreline changes observed along transects T1 to T16, the profile P2 refers to transects T17 to T32 and so on. For each Cell, using the year 1992 as a proxy shoreline, the Pearson's correlation matrix has been calculated to attempt a best fit and compare the temporal variations along the profiles. In addition, also the Student's t-test coefficient, $p$, has been

10   computed to investigate on the relevance of the correlation between the profiles, which has been assumed significant for $p$-values $< 0.05$. For the sake of brevity, the matrix of Student's t-test coefficient, $p$, has been omitted, but significant values of the correlation (characterized by $p<0.05$) have been written in Italics in the Pearson's correlation matrixes of Table 1, 2 and 3, respectively written for Cell I, Cell II and Cell III.

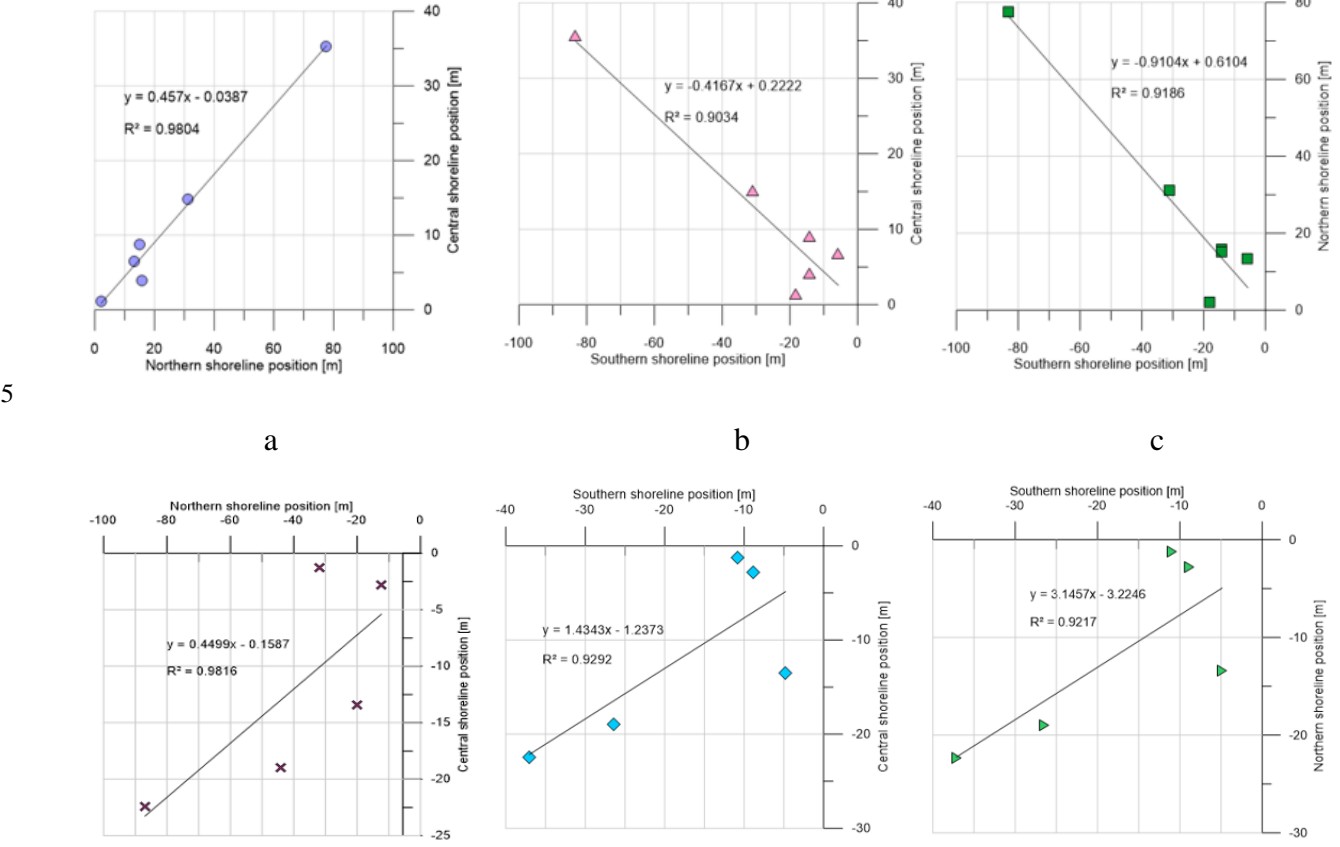

a                    b                    c



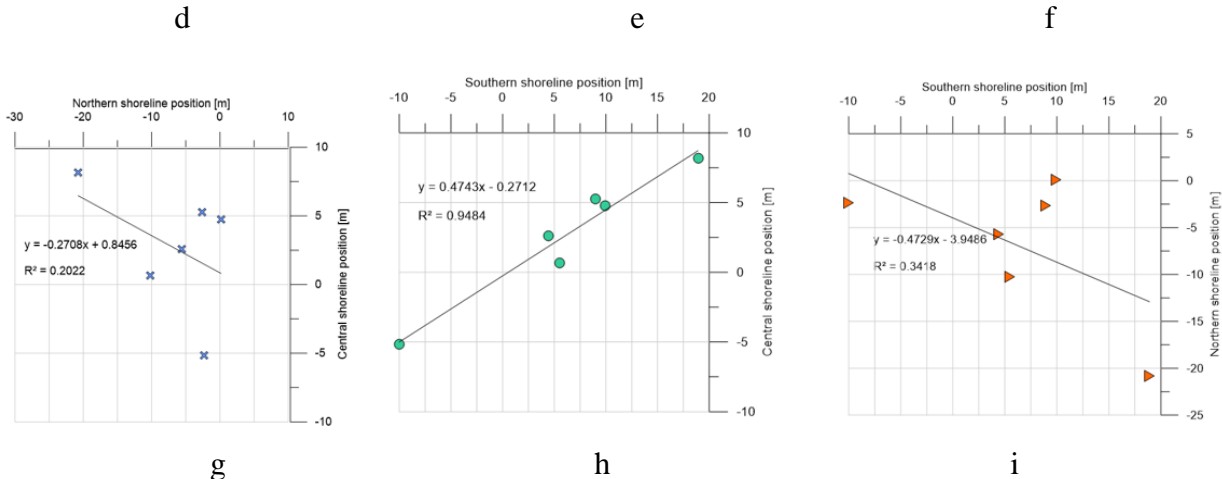

**Figure 13: Spatial correlations of the averaged shoreline changes during the investigated period between: northern and central shoreline; southern and central shoreline; southern and northern region in Cell I (a), (b), (c), Cell II (d), (e), (f) and Cell III (g), (h), (i).**

Each cell of the matrix displays the Pearson's correlation coefficient $r$ computed between the corresponding profiles on the first vertical column and on the first bottom row. The value of $r$ can range from -1 to 1. The sign indicates the direction of the relationship (that is, negative values imply an inverse relationship or a decreasing trend), while the absolute value indicates its strength, with larger (absolute) values meaning stronger linear relationships. The value $r = 0$ means absence of a linear relationship, even if other types of not linear relationships could relate the variables in any way. In Table 1, 2 and 3 positive correlations are colored in red and negative ones in blue.

In Table 1 significant positive relationships exist between the northern profiles in the range [P2–P8], so that when changes occur at one profile location they also occur on adjacent profiles. Specifically, the highest correlation is noted between P7 and P5 profiles ($r$=1.00). A similar scenario with positive correlations (generally moderate and high) is observed within the southern profiles [P13–P10]. The central profiles show both positive and negative correlations statistically irrelevant. From negligible to moderate are also correlations between the remaining central profiles and both southern and northern profiles. The statistically high and even very high negative correlations are of uttermost interest, expressing reliable inverse relationships. The high negative correlation ($r = –0.94$) between profiles P15 (extreme South) and P1 (extreme North) is stimulating as it proves the opposite trends in accretion/erosion patterns of northern and southern limits of Cell I, thus confirming the beach rotation resulting from the regression model. A negative correlation is also observed between the south and central sectors. Where the correlation signs change within the central region (turning from profile P9 to profile P10) a fulcrum is detected, i.e. the center of the beach acts as the axis of rotation, which is consistent with observation data, corresponding to a point distant around 4 km from North.





| | P1 | P2 | P3 | P4 | P5 | P6 | P7 | P8 | P9 | P10 | P11 | P12 | P13 | P14 |
|---|---|---|---|---|---|---|---|---|---|---|---|---|---|---|
| P2 | 0.95 | | | | | | | | | | | | | |
| P3 | 0.76 | 0.89 | | | | | | | | | | | | |
| P4 | 0.86 | 0.86 | 0.90 | | | | | | | | | | | |
| P5 | 0.77 | 0.86 | 0.97 | 0.97 | | | | | | | | | | |
| P6 | 0.78 | 0.91 | 0.98 | 0.91 | 0.97 | | | | | | | | | |
| P7 | 0.77 | 0.88 | 0.98 | 0.95 | 1.00 | 0.99 | | | | | | | | |
| P8 | 0.80 | 0.67 | 0.62 | 0.90 | 0.76 | 0.64 | 0.72 | | | | | | | |
| P9 | 0.34 | -0.26 | 0.17 | 0.54 | 0.37 | 0.15 | 0.29 | -0.83 | | | | | | |
| P10 | -0.84 | -0.47 | -0.37 | -0.68 | -0.48 | -0.43 | -0.47 | -0.87 | -0.61 | | | | | |
| P11 | -0.21 | -0.10 | 0.06 | -0.36 | -0.17 | 0.03 | -0.10 | -0.71 | -0.94 | 0.62 | | | | |
| P12 | 0.05 | 0.17 | 0.29 | -0.10 | 0.09 | 0.31 | 0.17 | -0.48 | -0.89 | 0.38 | 0.92 | | | |
| P13 | -0.74 | -0.24 | -0.17 | -0.49 | -0.27 | -0.21 | -0.24 | -0.75 | -0.60 | 0.96 | 0.62 | 0.46 | | |
| P14 | -0.87 | -0.38 | -0.35 | -0.54 | -0.37 | -0.40 | -0.38 | -0.64 | -0.28 | 0.91 | 0.28 | 0.06 | 0.91 | |
| P15 | -0.94 | -0.57 | -0.54 | -0.75 | -0.59 | -0.56 | -0.57 | -0.83 | -0.51 | 0.94 | 0.42 | 0.21 | 0.91 | 0.94 |

**Table 1: Pearson's correlation matrix applied to Cell I. Note: red marks positive relationships between profiles and blue marks negative relationships. Italic indicates $p<0.05$.**

| | P1 | P2 | P3 | P4 | P5 | P6 | P7 | P8 |
|---|---|---|---|---|---|---|---|---|
| P2 | 0,94 | | | | | | | |
| P3 | 0,78 | 0,86 | | | | | | |
| P4 | 0,88 | 0,92 | 0,93 | | | | | |
| P5 | 0,95 | 0,97 | 0,80 | 0,80 | | | | |
| P6 | 0,66 | 0,80 | 0,77 | 0,77 | 0,77 | | | |
| P7 | 0,43 | 0,65 | 0,87 | 0,87 | 0,87 | 0,74 | | |
| P8 | 0,62 | 0,70 | 0,91 | 0,91 | 0,91 | 0,85 | 0,82 | |
| P9 | 0,61 | 0,53 | 0,81 | 0,73 | 0,51 | 0,42 | 0,56 | 0,81 |

**Table 2: Pearson's correlation matrix applied to Cell II. Note: red marks positive relationships between profiles and blue marks negative relationships. Italic indicates $p<0.05$.**

| | P1 | P2 | P3 | P4 | P5 | P6 | P7 | P8 | P9 | P10 | P11 |
|---|---|---|---|---|---|---|---|---|---|---|---|
| P2 | 0,90 | | | | | | | | | | |
| P3 | 0.83 | 0.65 | | | | | | | | | |
| P4 | 0.78 | 0,80 | 0.65 | | | | | | | | |
| P5 | 0.36 | 0.28 | 0.79 | 0.61 | | | | | | | |
| P6 | -0.27 | -0,38 | 0,02 | 0.16 | 0.32 | | | | | | |
| P7 | 0.10 | -0,18 | 0,55 | 0,22 | 0,74 | 0.76 | | | | | |
| P8 | 0,22 | -0,11 | 0,57 | 0,22 | 0,63 | 0,75 | 0,97 | | | | |
| P9 | -0,27 | -0,04 | 0,55 | 0,36 | 0,66 | 0.83 | 0,96 | 0,98 | | | |
| P10 | -0.45 | -0,55 | -0,25 | 0.06 | 0.48 | 0.80 | 0,94 | 0.98 | 0,95 | | |
| P11 | -0,78 | -0,68 | -0,18 | -0.02 | 0.58 | 0.73 | 0,97 | 0,96 | 0,91 | 0,97 | |
| P12 | -0,92 | -0,64 | -0.71 | -0.12 | 0.36 | 0.92 | 0,80 | 0,73 | 0,75 | 0,79 | 0,82 |

**Table 3: Pearson's correlation matrix applied to Cell III. Note: red marks positive relationships between profiles and blue marks negative relationships. Italic indicates $p<0.05$.**





Table 2 shows mainly positive high correlations in Cell II, in northern, southern and central sectors, thus indicating that when changes take place at one profile location they also occur on the adjacent profiles. In Cell II no rotation is experienced, rather an almost uniform linear trend is noted, confirming the previous analysis.

In Cell III (Table 3), the highest positive values of *r* are noted for the profile couples [P3-P1], [P4-P2] in the northernmost area and [P8-P7], [P9-P8] in the southernmost part of the Cell III, indicating a concurrent trend in the coupled profiles when advance/retraction occurs. The highest negative values of *r* are observed for the profile couples [P11-P1], [P12-P1]. This means that when advance/retraction occurs in the southern region of the Cell, the opposite occurs in the northern one, still indicating a beach rotation. The fulcrum in Cell III is not so evident as in Cell I, where the tendency to rotation was more noticeable.

## 5 Numerical analysis results and discussion

All the three simulations executed with reference to the years 2008, 2011 and 2013 have been initialized with the bathymetry of the year 2006. Simulation S1 has used the initial coastline of the year 2005 and has run until year 2008, to compare the output coastline with 2008's observations. Simulation S2 has used the initial coastline of the year 2008 and has run until 2011, to compare with 2011's data. Simulation S3 has used the initial coastline of the year 2011 and has run until 2013, to compare with 2013's observations. It is worth mentioning that, since the end of 2015, submerged barriers have been built in the study area. These structures have not been included in the modelling, hence the simulations show the evolution of the coastline disregarding the possible effects of the abovementioned works.

The comparison between GIS results and numerical results for S1, S2 and S3 simulations is shown in Figure 14a, b, c for Cell I, Cell II and Cell III, respectively. For each Cell, the average of the observed and modelled shoreline variations in the sector (northern, central, southern) is displayed. The relative error, computed as the difference between the modelled and the measured value rated by the measured one, is also shown. Overall, the model seems to reasonably reproduce the observed erosion/accretion rates. Specifically, a quite good agreement is noted in Cell I and Cell III, while greater errors affect Cell II, especially in the northern sector, where the Ofanto's mouth is located.

In Cell I the computed error (in absolute value) is maximum in the northern sector for S2 run (28.3%) and minimum in the southern sector for S1 run (9.72%). For all runs, greatest errors are evident in the northern sector. The sign of the error highlights that the model always underestimates both the accretion of the shore occurring in the northern and central sectors of Cell I and overestimates the erosion occurring in the southern sector. In any case, a greater inaccuracy is noted in the estimation of the shoreline advance.



## Cell I

| | Northern (Average T1-T81) | | | Central (Average T82-T162) | | | Southern (Average T163-T244) | | |
|---|---|---|---|---|---|---|---|---|---|
| | Obs. | model | diff.(%) | Obs. | model | diff.(%) | Obs. | model | diff.(%) |
| ■ S1 (2005 - 2008) | 13,38 | 10,57 | -21,00% | 6,56 | 4,99 | -23,93% | -5,76 | -6,32 | 9,72% |
| ■ S2 (2008 - 2011) | 2,12 | 1,52 | -28,30% | 1,14 | 0,92 | -19,30% | -18,14 | -20,02 | 10,36% |
| ■ S3 (2011 - 2013) | 15,14 | 12,4 | -18,10% | 8,85 | 6,89 | -22,15% | -14,13 | -16,11 | 14,01% |

(a)

## Cell II

| | Northern (Average T1-T81) | | | Central (Average T82-T162) | | | Southern (Average T163-T244) | | |
|---|---|---|---|---|---|---|---|---|---|
| | Obs. | model | diff.(%) | Obs. | model | diff.(%) | Obs. | model | diff.(%) |
| ■ S1 (2005 - 2008) | -45,23 | -31,9 | -29,47% | -13,02 | -10,89 | -16,36% | 3,25 | 1,25 | -61,54% |
| ■ S2 (2008 - 2011) | -193 | -19,85 | -89,72% | -6,32 | -4,81 | -23,89% | -19,25 | -13,46 | -30,08% |
| ■ S3 (2011 - 2013) | -35,33 | -12,37 | -64,99% | -12,32 | -8,88 | -27,92% | -6,25 | -2,79 | -55,36% |

(b)




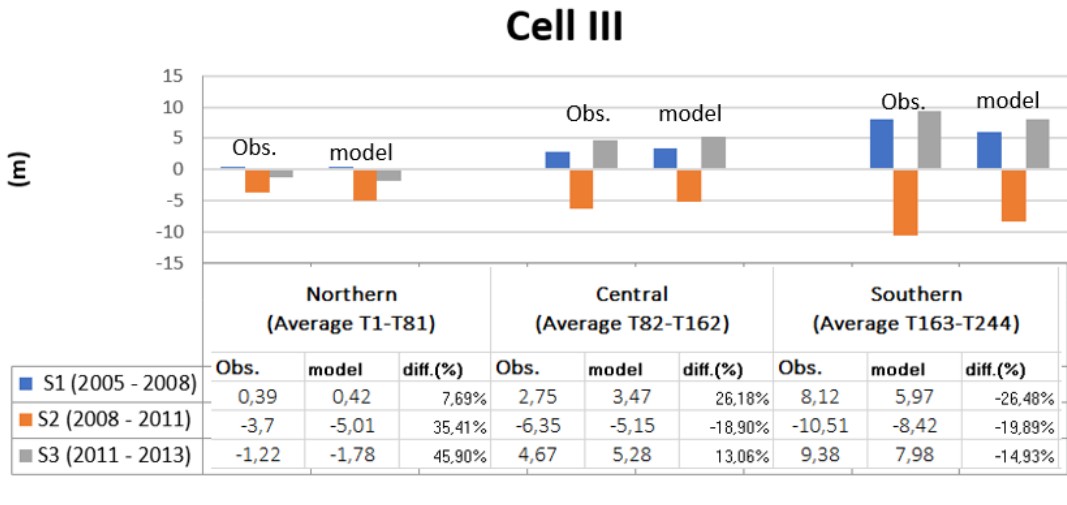

(c)

**Figure 14: Comparison between observation and model data a) Cell I, b) Cell II and c) Cell III. Negative values depict shoreline retreat, positive values shoreline advance (in meters).**

In Cell III the computed error (in absolute value) is maximum in the northern sector for S3 run (45.9%) and minimum in the same sector for S1 run (7.69%). In this case, for all runs, the model overestimates both accretion and erosion in the northern sector, while it underestimates both accretion and erosion in the southern sector. Comparing this behavior of the model with that observed in the simulations of Cell I, we note that the errors trend is not linear and depends on local effects.

This is even more evident when analyzing the computed error (in absolute value) in Cell II. It reaches 89.72% in the northern sector for S2 run, while its minimum value is 16.32% in the central sector for S1 run. A general underestimation of modelled values is noted in the whole Cell II in reproducing the shoreline retreat.

It is worth noting that in any case the model provides mean errors smaller than or equal to those reported in the literature for more complex models, such as multivariate linear regression models or evolutionary polynomial regression models (Goncalves

et al., 2012; Bruno et al., 2018). Based on Fig. 14a and 14c we can state that the model suitably allows the study of the coastline evolution in the case of a slightly curved shape beach profile. In Cell II, the 2D effects on the nearshore hydrodynamics and morphodynamics are so much relevant that they cannot be accurately modelled by a one-line model, such as the LITPACK.

Once recognized the limit of the model in the reproduction of the coastline, we have used it to attempt a prediction of the shoreline evolution from 2013 up 2018. The results are displayed in Figure 15 for Cell I, Cell II and Cell III, respectively,

providing the following information. In Cell I, the predicted accretion of the shoreline in the northern area is equal to 25.26 m on average and in the central area is equal to 15.43 m on average. In the southern area, erosion is predicted equal to 21.69 m on average. In Cell II, the model predicts an average erosion equal to 24.61 m in the norther area, to 16.65 m in the central area and to 10.56 m in the southern area. In Cell III, the predicted accretion of the shoreline in the southern area is 18.56 m on





average, while in the central area it is 10.19 m on average. In the northern area of Cell III, an almost stable shoreline is predicted (estimated average erosion of 1.51m).

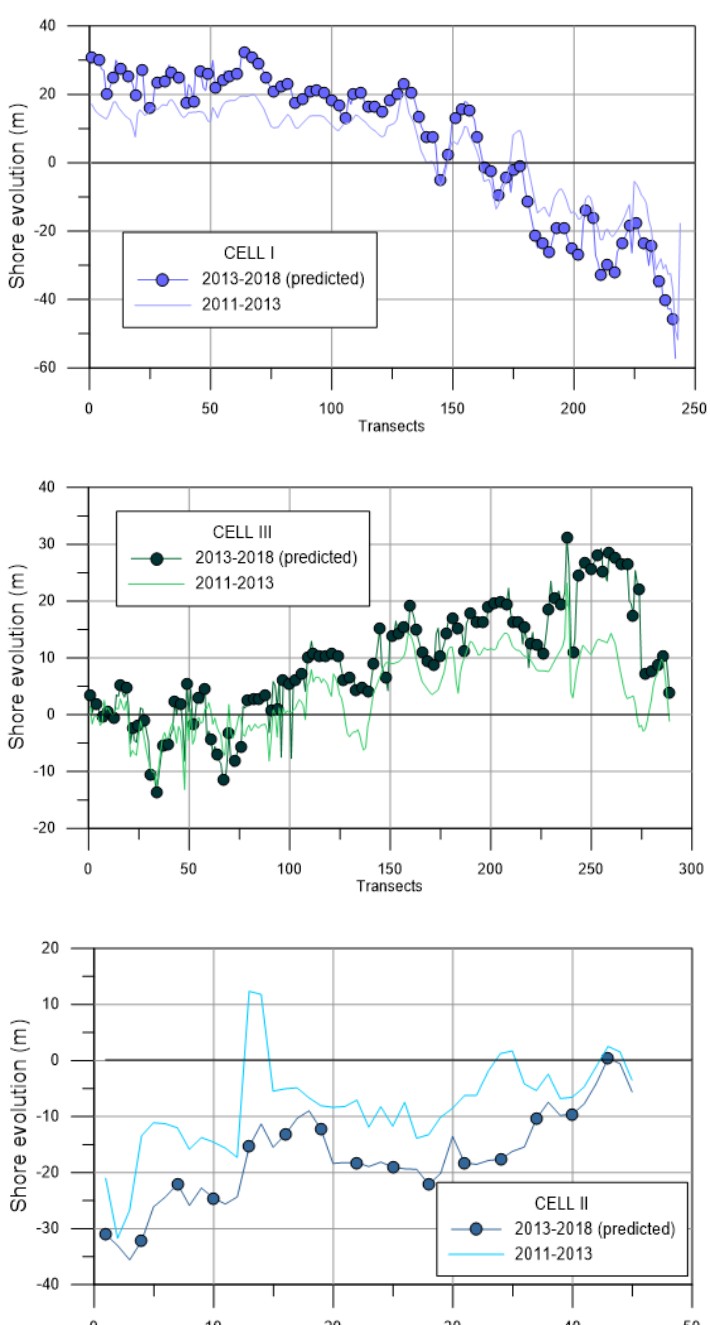

**Figure 15: Observed shoreline in the period 2011-2013 and predicted shoreline for the period 2013-2018 in (a) Cell I, (b) Cell II and c) Cell III.**



What is clear from this forecasting run is that the study area is in evolution and the beach equilibrium has not been achieved, yet. Specifically, the clockwise beach rotation already observed in Cell I and the anticlockwise beach rotation noted in Cell III are still expected also in the simulated period 2013-2018. In Cell I and Cell III the coastline evolves while maintaining its

concave shape. Furtherly, the changes in shoreline advance/regression have the same order of magnitude of those already analyzed in previous periods.

**6 Conclusion**

The present study has described an approach for the assessment of beach accretion/erosion, based on the joint use of data analysis, statistical methods and one-line numerical modelling. About 18 km of the southern Adriatic coast, showing two

concave beaches separated by a convex one, have been examined in the period 1992÷2013.

The temporal analysis of the shoreline variation by means of the GIS application has clearly shown the location of accretion and erosion areas. It has proved that in Cell I and Cell III the coastline has evolved, keeping its concave shape but rotating. A clockwise rotation has been observed in Cell I, with the formation of a northern area of sediments deposit and a southern erosion area. In Cell III an anticlockwise rotation of the coastline has produced an advance of the beach in the southern region

and a retreat in the northern one. Cell II has been characterized by a progressive erosion so that the convex shape beach profile has reduced over the years. These results have also been proved by the application of the linear regression model in each Cell and the computation of the Pearson's matrix, which have allowed to thoroughly investigate on correlations between northern, central and southern shoreline position.

These data have been used to validate the numerical one-line LITPACK model, specifically in the analysis of the shoreline in

the periods 2005-2008, 2008-2011 and 2011-2013. We have noted that the model is suitable in reproducing the shoreline evolution with a satisfactory accuracy in the case of slightly curved shape beaches (Cell I and Cell III) while greater errors have been obtained in all runs reproducing the shoreline evolution in Cell II, due to effects not handled by the model. Even if affected by this limitation, the model has finally been used to attempt a prediction for the period 2013÷2018. The result has shown that the shoreline has not reached an equilibrium yet and that the tendency already remarked in Cell I and Cell III (i.e.

clockwise and anticlockwise rotation, respectively) is confirmed also in predictive terms.

The proposed procedure has shown that this joint approach in the analysis of the coastline evolution is successful, providing complete information, both qualitative and quantitative, to stakeholders and identifying areas of erosion and deposition.

**Author contribution**

Elvira Armenio and Francesca De Serio prepared the manuscript with contributions from all co-authors.




**Competing interests**

The authors declare that they have no conflict of interest.

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
