# Peer review of "Coastline evolution based on statistical analysis and modelling"

_Natural Hazards and Earth System Sciences, 2018_

## Short Comment (SC1) · 4 Jan 2019

I read with interest the manuscript and found important insights on the shoreline dynamics in a semi-enclosed basin like the Adriatic Sea. Useful predictions have also been obtained by means of a simple numerical solver.

When speaking about the sea-level decrease occurred in 2011-2013 (P9, L5-6), the authors should better explain if the slight accretion also applies to Cells II and III. Further, such behavior could be compared with the shoreline evolution occurred a bit far from the investigated site (e.g., a general retreat has been recorded in that period for the beach of Senigallia, central Adriatic; see Postacchini, M., Soldini, L., Lorenzoni, C., & Mancinelli, A., 2017. "Medium-term dynamics of a middle Adriatic barred beach".

Ocean Science, 13, 5, 719-734).

For figures 4, 7 and 9, use of different line styles is suggested, to improve the readability of the shoreline change occurring over time.

When discussing the correlation between southern and northern portions of each cell (P15 and fig.12), it should be clarified if this refers to the only 1992-2013 curve.

References should be checked, as someone is missing.

---

## Referee Comment (RC1) · Anonymous Referee #1 · 22 Jan 2019

General comments

The paper concerns the estimation of coastline evolution due to erosion and/or deposition processes. In particular Authors present an approach to analyse the dynamics of a coastal system, identifying typical and recurrent erosion/accretion processes. The presented approach is based on the combination of field information, GIS tools and statistical models applications and is tested on a coastal reach in the southern Italy. The subject is of strong interest and the presented material is technically good. In my opinion the paper is acceptable but some elements should be improved to better highlight the contribution of the work and the advantages of the proposed procedure.

Specific questions to consider: - In Introduction section (on page 1): In this part of the work I suggest to introduce some comments on the recent research on the

dynamics evolution in transitional environments as the coastal aereas (a brief review can be found in the recent Special volume of Advances in Water Resources (2018), Volume 120). As an example it could be interesting to highlight how the dynamism of coastal areas could result both from the fluvial system dynamism and from sea-level dynamism. - Results section: in my opinion the authors should better explain the behaviour in cell I, Cell II and Cell III. In my opinion a figure which more clearly shows the accretion and retreat for each cell could be introduced (as an example from figures 7 and 8 it is not easy to highlight it). - Numerical analysis results and discussion section: I suggest to introduce more details and discussion about error estimation and uncertainty of the results.

Please also note the supplement to this comment:
https://www.nat-hazards-earth-syst-sci-discuss.net/nhess-2018-239/nhess-2018-239-RC1-supplement.pdf

---

## Referee Comment (RC2) · Anonymous Referee #2 · 9 Jul 2019

The manuscript describes an interannual analysis of the shoreline evolution in a dynamic coastal area in South Italy by means of field observations, statistical tools and 1D commercial model. The topic is certainly of interest for the readers of NHESS; however, some flaws characterize the overall description of the adopted methodologies and results and my recommendation is to accept the manuscript for publication pending major revisions, mainly concerning some clarification on the adopted methodologies, as noted in the following comments.

- In Section 1, the Authors are suggested to review the new integrated approach proposed to assess coastal vulnerability to beach changes (among the others, Bonaldo et al. (2019) Integrating multidisciplinary instruments for assessing coastal vulnerability to erosion and sea level rise: lessons and challenges from the Adriatic Sea, Italy. Journal

of Coastal Conservation, 23 (1), pp. 19-37) - In Section 2, the Authors are suggested to add information on the morphological features of the area, such as the longshore transport in the area (main direction and rate), on the (if available) solid discharge from Ofanto river, closure depth. - In Section 2, the Authors are suggested to review Figure 1. The study area extends from the harbor of Margherita di Savoia to Barletta, and not including Gulf of Manfredonia. - In Section 3.1, the Authors are suggested to specify the sources of the analyzed aerial photography images and of the transects. - In Section 3.2, the y-labels in Figures 6, 8 and 10 should be correct into m and not m/year, if I well understood. - In Section 3.3, in the linear regression model (Figure 13), the Authors are suggested to clarify the definition of x variable and its calculation (i.e., central position of each section starting from the the northern one). - In Section 3.3, the Authors are suggested to clarify the input conditions for waves, wind, water levels. - In Figure 13, the caption text misses some years reported in the graph and in the legend. - The presentation and quality of Figure 14 are suggested to be improved. - The new grouping of the transects as shown in Tables on page 20 is a little bit confusing in reference to the results previously described. The Authors are suggested to improve the presentation of this analysis. - In Section 5, the Authors are suggested to discuss the feasibility to run simulations for longer periods (i.e., 2005 - 2013 or shorter period 2005-2011 and 2008-2013) and eventually compare the computational results with observations. - A review of the English language is also suggested.

---

## Author Response (AR1)

I read with interest the manuscript and found important insights on the shoreline dynamics in a semi-enclosed basin like the Adriatic Sea. Useful predictions have also been obtained by means of a simple numerical solver.

When speaking about the sea-level decrease occurred in 2011-2013 (P9, L5-6), the authors should better explain if the slight accretion also applies to Cells II and III. Further, such behavior could be compared with the shoreline evolution occurred a bit far from the investigated site (e.g., a general retreat has been recorded in that period for the beach of Senigallia, central Adriatic; see Postacchini, M., Soldini, L., Lorenzoni, C., & Mancinelli, A., 2017. "Medium-term dynamics of a middle Adriatic barred beach".

Ocean Science, 13, 5, 719-734).

For figures 4, 7 and 9, use of different line styles is suggested, to improve the readability of the shoreline change occurring over time.

When discussing the correlation between southern and northern portions of each cell (P15 and fig.12), it should be clarified if this refers to the only 1992-2013 curve.

References should be checked, as someone is missing.

[Figure]

We would like to thank you for the positive comments on our paper and for your useful suggestions, opening a fruitful discussion phase. Following your hints, we have added a brief part in the paragraph "Statistical analysis results and discussion" (P9, L5-6), to better explain the slight accretion which occurs in the investigated areas. As suggested, referring to the paper "Medium-term dynamics of a middle Adriatic barred beach" (Postacchini, M. et al., 2017), we have reported a comparison with the shoreline evolution occurred of the beach of Senigallia in the central Adriatic Sea.

We have also modified figures 4, 7 and 9, by increasing the different lines width to improve the readability of the shoreline change over time. Different line styles were not

applicable because of lines overlapping (making even more confusing the plot).

Finally, we have inserted a comment on the correlation between southern and northern portions of each cell (P15 and fig.12), specifying that it refers only to the 1992-2013 curve.

References have been checked, thanks.

Please also note the supplement to this comment:
https://www.nat-hazards-earth-syst-sci-discuss.net/nhess-2018-239/nhess-2018-239-AC1-supplement.pdf

Nat. Hazards Earth Syst. Sci. Discuss.,
https://doi.org/10.5194/nhess-2018-239-RC1, 2019

[Figure]

The paper concerns the estimation of coastline evolution due to erosion and/or deposition processes. In particular Authors present an approach to analyse the dynamics of a coastal system, identifying typical and recurrent erosion/accretion processes. The presented approach is based on the combination of field information, GIS tools and statistical models applications and is tested on a coastal reach in the southern Italy. The subject is of strong interest and the presented material is technically good. In my opinion the paper is acceptable but some elements should be improved to better highlight the contribution of the work and the advantages of the proposed procedure.

Specific questions to consider: - In Introduction section (on page 1): In this part of the work I suggest to introduce some comments on the recent research on the

dynamics evolution in transitional environments as the coastal aereas (a brief review can be found in the recent Special volume of Advances in Water Resources (2018), Volume 120). As an example it could be interesting to highlight how the dynamism of coastal areas could result both from the fluvial system dynamism and from sea-level dynamism. - Results section: in my opinion the authors should better explain the behaviour in cell I, Cell II and Cell III. In my opinion a figure which more clearly shows the accretion and retreat for each cell could be introduced (as an example from figures 7 and 8 it is not easy to highlight it). - Numerical analysis results and discussion section: I suggest to introduce more details and discussion about error estimation and uncertainty of the results.

Please also note the supplement to this comment:
https://www.nat-hazards-earth-syst-sci-discuss.net/nhess-2018-239/nhess-2018-239-RC1-supplement.pdf

Nat. Hazards Earth Syst. Sci. Discuss.,
https://doi.org/10.5194/nhess-2018-239-AC2, 2019

[Figure]

We would like to thank you for your positive comments and helpful suggestions about the modifications necessary for the revision of our paper. All the changes made in the paper are reported in green color. Following your hints, we have added, in the Introduction section (on page 1), some comments with reference to the recent research on the dynamics evolution in transitional environments as the coastal areas. In particular, we referred to the paper "River processes and links between fluvial and coastal systems in a changing climate", Advances in Water Resources (Termini, 2018). In the Result section, according to your suggestion, we have added a map (Figure 4 on page 9) which more clearly shows the accretion and retreat areas in Cell I, Cell II and Cell III during the observation period. Finally, in the Numerical analysis results

and discussion section, we have added a comment, on page 21, to better explain the computation of the relative error with reference to averaged transects for the northern, the central and the southern sector.

Please also note the supplement to this comment:
https://www.nat-hazards-earth-syst-sci-discuss.net/nhess-2018-239/nhess-2018-239-AC2-supplement.pdf

Nat. Hazards Earth Syst. Sci. Discuss.,
https://doi.org/10.5194/nhess-2018-239-RC2, 2019

[Figure]

The manuscript describes an interannual analysis of the shoreline evolution in a dynamic coastal area in South Italy by means of field observations, statistical tools and 1D commercial model. The topic is certainly of interest for the readers of NHESS; however, some flaws characterize the overall description of the adopted methodologies and results and my recommendation is to accept the manuscript for publication pending major revisions, mainly concerning some clarification on the adopted methodologies, as noted in the following comments.

- In Section 1, the Authors are suggested to review the new integrated approach proposed to assess coastal vulnerability to beach changes (among the others, Bonaldo et al. (2019) Integrating multidisciplinary instruments for assessing coastal vulnerability to erosion and sea level rise: lessons and challenges from the Adriatic Sea, Italy. Journal

of Coastal Conservation, 23 (1), pp. 19-37) - In Section 2, the Authors are suggested to add information on the morphological features of the area, such as the longshore transport in the area (main direction and rate), on the (if available) solid discharge from Ofanto river, closure depth. - In Section 2, the Authors are suggested to review Figure 1. The study area extends from the harbor of Margherita di Savoia to Barletta, and not including Gulf of Manfredonia. - In Section 3.1, the Authors are suggested to specify the sources of the analyzed aerial photography images and of the transects. - In Section 3.2, the y-labels in Figures 6, 8 and 10 should be correct into m and not m/year, if I well understood. - In Section 3.3, in the linear regression model (Figure 13), the Authors are suggested to clarify the definition of x variable and its calculation (i.e., central position of each section starting from the the northern one). - In Section 3.3, the Authors are suggested to clarify the input conditions for waves, wind, water levels. - In Figure 13, the caption text misses some years reported in the graph and in the legend. - The presentation and quality of Figure 14 are suggested to be improved. - The new grouping of the transects as shown in Tables on page 20 is a little bit confusing in reference to the results previously described. The Authors are suggested to improve the presentation of this analysis. - In Section 5, the Authors are suggested to discuss the feasibility to run simulations for longer periods (i.e., 2005 - 2013 or shorter period 2005-2011 and 2008-2013) and eventually compare the computational results with observations. - A review of the English language is also suggested.

Nat. Hazards Earth Syst. Sci. Discuss.,
https://doi.org/10.5194/nhess-2018-239-AC3, 2019

[Figure]

The manuscript describes an interannual analysis of the shoreline evolution in a dynamic coastal area in South Italy by means of field observations, statistical tools and 1D commercial model. The topic is certainly of interest for the readers of NHESS; however, some flaws characterize the overall description of the adopted methodologies and results and my recommendation is to accept the manuscript for publication pending major revisions, mainly concerning some clarification on the adopted methodologies, as noted in the following comments. We would like to thank Referee#2 for her/his detailed revision work that will help us to improve our manuscript.
* * *
- In Section 1, the Authors are suggested to review the new integrated approach proposed to assess coastal vulnerability to beach changes (among the others, Bonaldo et al. (2019) Integrating multidisciplinary instruments for assessing coastal vulnerability to erosion and sea level rise: lessons and challenges from the Adriatic Sea, Italy. Journal of Coastal Conservation, 23 (1), pp. 19-37).

Thanks for this suggestion. We read the paper, which is very interesting, and we have also cited it in this revised version.

- In Section 2, the Authors are suggested to add information on the morphological features of the area, such as the longshore transport in the area (main direction and rate), on the (if available) solid discharge from Ofanto river, closure depth.

The prevailing direction of the solid longitudinal transport along the Apulian coast is from north to south. At the moment, other information is not available.

- In Section 2, the Authors are suggested to review Figure 1. The study area extends from the harbor of Margherita di Savoia to Barletta, and not including Gulf of Manfredonia.

Thanks, done.

- In Section 3.1, the Authors are suggested to specify the sources of the analyzed aerial photography images and of the transects.

The aerial photographs were directly acquired by the Research Unit of our Dpt. during some surveys campaign. This information has been added in section 3.1.

- In Section 3.2, the y-labels in Figures 6, 8 and 10 should be correct into m and not m/year, if I well understood.

Thanks, there were some misprints in these Figures and have been corrected.

- In Section 3.3, in the linear regression model (Figure 13), the Authors are suggested to clarify the definition of x variable and its calculation (i.e., central position of each

section starting from the northern one).

Thanks, we have modified the Figures 13a, 13b, and 13c and we have clarified the definition of x variable and its calculation.

- In Section 3.3, the Authors are suggested to clarify the input conditions for waves, wind, water levels.

The input conditions for the LITPACK were already described in 3.3, but some other notes have been added in this revised version.

- In Figure 13, the caption text misses some years reported in the graph and in the legend.

Thanks, done.

- The presentation and quality of Figure 14 are suggested to be improved.

Thanks, done.

- The new grouping of the transects as shown in Tables on page 20 is a little bit confusing in reference to the results previously described. The Authors are suggested to improve the presentation of this analysis.

The grouping of the numerous transects into profiles was necessary to compute the Pearson coefficients and allow readable Tables. The way in which this grouping was carried out was already described in the paper.

- In Section 5, the Authors are suggested to discuss the feasibility to run simulations for longer periods (i.e., 2005 - 2013 or shorter period 2005-2011 and 2008-2013) and eventually compare the computational results with observations.

Actually, the simulation was carried out in the described way, in order to validate the model in similar temporal periods. Once validated, we had used the validated coastline as the new input data for the successive run. In this way, there was a multiple check

on the model performance.

- A review of the English language is also suggested.

Thanks, we have revised the English language.

Please find attached as supplement the revised paper.

Please also note the supplement to this comment:
https://www.nat-hazards-earth-syst-sci-discuss.net/nhess-2018-239/nhess-2018-239-AC3-supplement.pdf

[revised manuscript text omitted]

**List of all relevant changes made in the manuscript**

- Introduction

- Section 1 and 2, in particular the Paragraph "Statistical analysis results and discussion"

- Figures 4, 6, 7, 8, 9, 10

- English check

- References